# Silicon/Graphite/Amorphous Carbon as Anode Materials for Lithium Secondary Batteries

**DOI:** 10.3390/molecules28020464

**Published:** 2023-01-04

**Authors:** Haojie Duan, Hongqiang Xu, Qian Wu, Lin Zhu, Yuting Zhang, Bo Yin, Haiyong He

**Affiliations:** 1School of Materials Science and Chemical Engineering, Ningbo University, Ningbo 315211, China; 2Ningbo Institute of Materials Technology and Engineering, Chinese Academy of Sciences, Ningbo 315201, China

**Keywords:** lithium-ion batteries, anode material, Si/G/C, spray drying

## Abstract

Although silicon is being researched as one of the most promising anode materials for future generation lithium-ion batteries owing to its greater theoretical capacity (3579 mAh g^−1^), its practical applicability is hampered by its worse rate properties and poor cycle performance. Herein, a silicon/graphite/amorphous carbon (Si/G/C) anode composite material has been successfully prepared by a facile spray-drying method followed by heating treatment, exhibiting excellent electrochemical performance compared with silicon/amorphous carbon (Si/C) in lithium-ion batteries. At 0.1 A g^−1^, the Si/G/C sample exhibits a high initial discharge capacity of 1886 mAh g^−1^, with a high initial coulombic efficiency of 90.18%, the composite can still deliver a high initial charge capacity of 800 mAh g^−1^ at 2 A g^−1^, and shows a superior cyclic and rate performance compared to the Si/C anode sample. This work provides a facile approach to synthesize Si/G/C composite for lithium-ion batteries and has proven that graphite replacing amorphous carbon can effectively improve the electrochemical performance, even using low-performance micrometer silicon and large size flake graphite.

## 1. Introduction

Rechargeable lithium-ion batteries (LIBs) were successfully commercialized by Sony in 1991 [1]. LIBs have many advantages, such as high energy density, good cycle stability, low self-discharge effect, no memory effect, and environmental friendliness, compared with nickel-metal hydride, nickel-cadmium, and lead-acid batteries. These advantages for lithium-ion batteries have increased their application areas, especially in the area of consumer electronics products [2]. However, with the rapidly increasing demand for higher energy density, the current theoretical capacity of graphite cathodes is too low to meet the requirements of further applications of LIBs [3]. Consequently, there is growing interest in developing battery electrodes with high gravimetric and volumetric capacity to surpass the energy density of the current LIBs [2]. Silicon is widely considered as one of the most promising anode materials. Nevertheless, the Si electrode suffers from large undesired capacity loss caused by the large volume expansion during the repeated cycling and the low electric conductivity, causing a short cycle life during application [4].

To address these critical issues, considerable strategies have been invented to improve the performance of Si anodes, such as optimizing electrode structure. Nanostructures have been demonstrated to be an effective method given the modification of conductivity as well as the alleviation of swelling volume [5]. For example, silicon nanowires can solve the material pulverization problem as they can accommodate large strains. Furthermore, they also provide good electronic contact and conduction, and shorten the lithium insertion distances [6]. In addition, it is also an effective technique for the integration of well-conductive species into Si, which may operate as a buffer to cushion structural fracture and improve electric connection.

Recently, various Si-based anodes have been developed to provide significantly greater lithium storage performances than the pure Si anode, such as Si/C [7,8,9,10], Si/Ag [11], Si/Sn [12,13], Si/Cu [14], and Si/conducting polymer composites [15,16]. Among these buffer metals, the co-utilization of silicon and graphite provides the most practical and easy to commercialize anode for high-energy lithium batteries [17]. Compared with silicon anode, graphite has more advantages, such as low cost, high CE, excellent cycle life, good mechanical flexibility, small volume change, and high conductivity. Therefore, to achieve high specific capacity, area capacity, and volume capacity, silicon is added to the graphite negative electrode, and at the same time to buffer volume changes and increases conductivity. In addition, the co-utilization of silicon and graphite can be achieved using the same commercial production line, thus translating into high manufacturability and minimal investment. Silicon–graphite composite will simultaneously maximize the advantages of both materials while decreasing the disadvantages of both and also ensuring its success as a viable alternative for the battery industry. However, integrating silicon and graphite into a single composite to obtain the desired properties remains a challenge, as the two materials differ significantly in terms of physical and chemical properties.

Herein, we report the preparation of Si/G/C composite anodes containing micro silicon particles, flake graphite, and amorphous carbon generated from sucrose. The low-cost micron silicon particles were obtained from the scraps of photovoltaic silicon production lines. This work attempts to mix these low-cost micron Si particles, flake graphite, and sucrose to prepare Si/G/C anode materials by a facile spray-drying and carbonization process. The graphite and amorphous carbon in the composite can act as a buffer matrix for the Li–Si alloy during the cycle. In addition, amorphous carbon can successfully link Si with graphite, generating a conductive network, and may efficiently contain the Si and graphite particles inside it. The Si/G/C composite obtained shows good electrochemical performance, compared with the Si/C composite.

## 2. Results and Discussion

The XRD curves of a pristine Si/G/C sample along with a control sample of Si/C are shown in Figure 1a. As to the Si/G/C and Si/C, a sequence of diffraction peaks can be recognized. The diffraction peaks at 2θ = 28.44°, 47.30°, 56.12°, 69.13°, 76.38°, and 88.03° are attributed to the (111), (220), (311), (400), (331), and (422) planes of the metal Si (PDF#75-0589), respectively. While the diffraction peaks at 2θ = 26.38°, 42.22°, 44.39°, 54.54°, 77.24°, and 83.18° correspond to the (002), (100), (101), (004), (110), and (112) planes of the graphite (PDF#41−1487). Noticeably, no apparent diffraction peaks of amorphous carbon can be noticed in the Si/C sample, perhaps owing to its low degree of crystallinity. All of the diffraction patterns of Si/G/C composites are perfectly consistent with Si and graphite, demonstrating that the crystalline phase change of Si does not take place during the spray drying and carbonization process. The Raman spectra of Si/G/C and Si/C are displayed in Figure 1b. The ratio of the integrated area of the D band and G band reveals the graphitization degree. Compared with Si/C, the lower ratio of ID/IG for Si/G/C shows that Si/G/C anodes offer greater electrical conductivity. Compared with graphite, Si/G/C and Si/C samples exist as amorphous sucrose-pyrolyzed carbon. All this evidence suggests that the Si/G/C and Si/C composites are successfully prepared.

The thermogravimetric (TG) curves of Si/G/C and Si/C composites, measured in air at 900 °C to burn away the carbon matrix, are shown in Figure 2. TG analysis indicates that Si/G/C powders contain ≈58.44 wt.% of Si and 41.56 wt.% of carbon. The experimental idea is that the G of Si/G/C is to replace part C in Si/C, the carbon residual rate of sucrose is about 33.33%, and 3 g sucrose is replaced by 1 g graphite. In fact, graphite also loses weight after heat treatment, resulting in less carbon.

Figure 3a,b display the SEM images of the micro-sized porous Si and the graphite. As demonstrated in Figure 3a, the silicon displays the particle size with micro-scale, without evident agglomeration. As indicated in Figure 3b, the particle size of flake graphite is about 15 μm. Figure 3c–f show the SEM images of Si/C precursor, Si/C, Si/G/C precursor, and Si/G/C composites, respectively. Although spherical particles may be seen in the precursors produced by spray drying, they cannot stay that way after being heated, perhaps because the D50 of Si and graphite are both micron-sized. There is no micron Si on the surface of the Si/C material, showing that micron silicon is encased in amorphous carbon. For the Si/G/C material, it is clear that the graphite sheet is bigger and not entirely covered. From Figure 3b,f, it is seen that the graphite experiences ferocious shattering making the particles smaller during ball milling. N_2_ adsorption/desorption isotherms of Si/G/C with graphite added may be categorized as type IV with H4 hysteresis loop in the relative pressure (p/p0) range between 0.5 and 1.0, which is associated with meso-mesopores (Appendix A). The morphological alterations brought on by significant volume variations may be reduced by the meso-macropore structure, which enhances cyclic stability [18,19,20].

According to the element mapping in Figure 4a, the Si and C elements are distributed equally throughout the composite, showing the components of the Si/G/C composite are evenly mixed. The amorphous carbon may efficiently enclose the Si and graphite particles inside it. It is stated that the inclusion of amorphous carbon on the surface of the composite would prevent the silicon particles from separating by providing effective constrain force and is favorable for the electrical transmission, thus, it may increase the electrochemical performance of the material. Figure 4b,c show the HRTEM pictures and the selected area electron diffraction (SAED) pattern of the composite, from which we can discern the main components of the composite.

Figure 5a depicts the typical galvanostatic charge–discharge curves of the Si/G/C and Si/C composite at the current density of 1 A g^−1^ within 0.01–2 V (vs. Li/Li^+^). The Si/G/C anode produces a high initial discharge capacity reaching 1886 mAh g^−1^ with an initial coulombic efficiency of 90.18% and irreversible capacity of 185.2 mAh g^−1^ as a consequence of the creation of solid electrolyte interface (SEI) [21]. The capacity will increase with the decreasing particle size of active materials, which is attributed to the reduced distance travelled by the lithium-ion in smaller particles during the intercalation process at a given C-rate and the greater proportion of edge sites and the increased utilization rate of smaller particles during the intercalation process (charge storage) [22].The discharge curves feature a lengthy and distinct platform at 0.2–0.01 V that may be related to the lithiation process of Si, graphite, and amorphous carbon. Additionally, the charge curves exhibit potential plateaus at around 0.5 V and the below are ascribed to the de-lithiation of Si and carbon, respectively. The findings will be further validated by the CV curves in Figure 5f. As shown in Figure 5c, the Si/G/C anodes display higher rate capability when the charge current density is adjusted from 0.1 A g^−1^ to 2 A g^−1^ (Figure 5b), even when the charge and discharge current density increased to 1 A g^−1^ to test the circulation properties (Figure 5d). As shown in Figure 5e, the Si/G/C anodes exhibit the high initial coulombic efficiency (90.18%) and average coulombic efficiency (99%) after about 30 cycles which are used as a rate performance test.

The CV curves of the Si/G/C composite within the voltage of 0.01–1.5 V are presented in Figure 5f. Figure 5f depicts the first five scanning curves at a scanning rate of 0.1 mV s^−1^, a visible reduction peak at 0.2 V, and the oxidation peaks at 0.32 V and 0.48 V may be ascribed to the silicon alloying/dealloying processes with Li^+^ [23]. During the initial cathodic scanning operation, the faint peak at 0.5 V only observed in the first cycle may be attributed to the creation of the SEI layer on the surface of the active materials. The strength of the peaks rapidly grew throughout the successive cycles, suggesting that the Si-based materials were increasingly activated. The CV profiles of the Si/C are similar to that of Si/G/C except that there is only one obvious oxidation peak (Appendix A).

In order to better understand the kinetics of lithium storage, CV curves for the Si/G/C and Si/C were obtained throughout a scan rate range of 0.2 to 1.0 mV s^−1^ in a voltage window of 0.01 to 1.5 V (Appendix A). Even at a fast sweep speed, it is clear that all of the samples’ CV curves, which include a cathodic peak and an anodic peak, are comparable. The response currents of the samples’ redox peaks grew quickly with the stepwise increase in scan speeds, although the peak locations hardly changed. Peak current (i) and scanning rate (v) have the following connection, according to the literature [24,25,26]: $i=a\;*\;v^{b}$, where a and b are the adjustable constants, and the b value may be derived using the slope of log(i)–log(v). The diffusion-controlled process or the capacitance-limited mechanism for lithium storage behavior, respectively, are indicated by a b value of 0.5 or 1.0. As can be shown in Appendix A, all sample b values were near to 0.5, demonstrating that diffusion behavior governed the processes of lithium storage in the produced electrode materials. To gain an insight into the lithium diffusion kinetics and electrical resistance of the samples, electrochemical impedance spectroscopy (EIS) measurements were carried out in the frequency range of 0.01 Hz to 1 MHz with an amplitude of 10 mV. The Si/G/C and Si/C Nyquist graphs are shown in Appendix A. Si/G/C has a considerably lower Rp value than Si/C, which implies that graphite could improve the connection between active materials and the current collector and increase the conductivity of anode materials. As a result, the phase-separated graphite significantly improves the electrochemical performance of the Si/G/C anode material. The lithium-ion diffusion coefficient was measured by GITT with a pulse current of 0.1 A g^−1^ for a pulse time of 30 min between 1 h rest intervals (Appendix A) and calculated by Equation [26]: $D=\frac{4L^{2}}{\pi \tau }{\left(\frac{\Delta E_{s}}{\Delta E_{\tau }}\right)}^{2}$. From Appendix A, it can be observed that the calculated D_Li_^+^ of Si/G/C for both the lithiation and de-lithiation processes are close to those of the Si/C anode, which agrees well with the calculation results from CV data (Appendix A).

## 3. Experimental Section

**Synthesis of Si/C and Si/G/C Materials.** Typically, 2 g of PVP (the molecular weight: 360,000) and 3 g of sucrose were successively dissolved in 60 mL of distilled water under stirring within 30 °C. Then, 2 g micro-Si and 1 g flake graphite were orderly dispersed into the above solution by vigorous stirring, followed by ball milling for 4 h to form a homogeneous suspension solution. The resulting solution was used to fabricate Si/G/C composites by the spray drying process. The obtained Si/G/C composites were annealed at 200 °C for 2 h with a heating rate of 1 °C min^−1^ under an air atmosphere, then calcined at 900 °C for 2 h in an argon atmosphere. After the calcination, the obtained Si/G/C composites were denoted as Si/G/C (Si/Graphite/Carbon). For comparison, Si/C (Si/Carbon) was also prepared under the same condition except that 1 g of graphite was replaced by an additional 3 g of sucrose.

**Structural characterization.** The X-ray diffraction (XRD) patterns were measured by X-ray diffractometer (D8 ADVANCE DAVINCI, Cu kα radiation, λ = 0.154 nm). S4800 cold field-emission scanning electron microscopy (SEM) and transmission electron microscopy (TEM Tecnai F20) were used to reveal the microstructure of the as-obtained samples. A Confocal Raman Reflectance Microscope (Ram Enishaw Invia REFLEX) was used to accurately analyze the crystallinity and defects of the sample. Thermogravimetric analysis (TGA) was carried out using TGA 8000-Spectrum two-Clarus SQ8T instrument at a heating rate of 10 °C min^−1^ from 20 °C to 900 °C under air. The adsorption data of the multipoint Brunauer–Emmett–Teller (BET) method were used to calculate the specific surface area and pore size of the sample.

**Preparation of anode electrode.** The obtained products served as electrode active material in half-cells for the study of their electrochemical lithium-storage performance. The assembling of the half-cell involved the following phases. First and foremost, making the electrode slurries, the active material (Si/G/C or Si/C), conductive carbon (Super P) and sodium carboxymethyl cellulose (CMC) binder with a mass ratio of 8:1:1 were mixed thoroughly and then transferred into the sample bottle, dissolved in the deionized water, and stirred for 6 h. Then, the working electrodes were fabricated by spreading the electrode slurries on Cu foil and dried at 80 °C overnight under vacuum conditions. Following that, the sheet-punching machine cut 12 mm diameter electrode slices, which were then pressed for 30 s at a pressure of 20 MPa. The mass loading of the active material was between 0.99 and 1.13 mg cm^−2^ (based on the quality of Si/G/C or Si/C) in the prepared electrode. Afterwards, the CR2032 type coin cells were assembled in argon-filled humidity-free glove box using pure lithium foils as a counter electrode with 1M LiPF6 dissolved in ethylene carbonate (EC)/diethyl Carbonate (DEC)/ethylene methyl carbonate (EMC) (1:1:1, volumetric ratio) as the electrolyte.

**Electrochemical measurements.** The multichannel battery test system (LAND CT-2001A; Wuhan Rambo Testing Equipment Co., Ltd., Wuhan, China) was applied to measure the galvanostatic in the voltage range from 0.01 to 2 V (vs. Li/Li^+^) and different current densities. Cyclic voltammetry (CV) curves and Electrochemical impedance spectroscopy (EIS) measures were obtained by using a Solartron 1470E (Solartron Analytical, UK) multi-channel potentiostat electrochemical workstation. All of the electrochemical experiments indicated above were carried out at room temperature.

## 4. Conclusions

In summary, Si/G/C anode materials were effectively manufactured by a straightforward spray-drying and annealing technique. Both silicon and graphite were covered with amorphous carbon. As a result, the Si/G/C anode displayed a superior cycling performance and a greater initial coulombic efficiency as compared to the Si/C composite. The high electrochemical performance was due to the covering of amorphous carbon, and specifically the buffering of graphite substrate. The production of Si/G/C presents an easy technique and unique microstructure which may be utilized for creating various anode materials on a large scale, and concurrently provides a viable foundation for the influence of graphite or sucrose carbon on the Si-based anodes in lithium-ion batteries. Therefore, it is a good prospect to fabricate Si/G/C with better properties of graphite and nano-silicon materials.

## Figures and Tables

**Figure 1 molecules-28-00464-f001:**
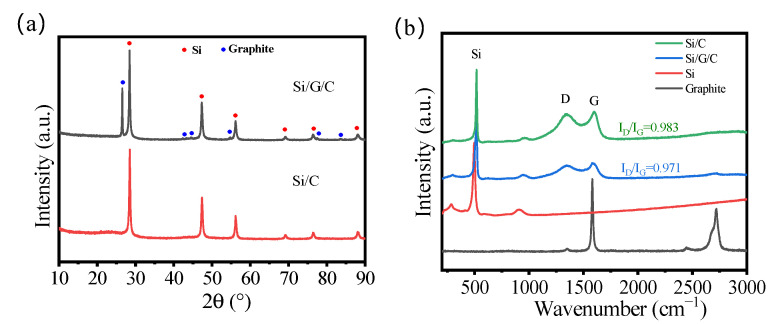
(**a**) XRD patterns; (**b**) Raman spectra of various samples.

**Figure 2 molecules-28-00464-f002:**
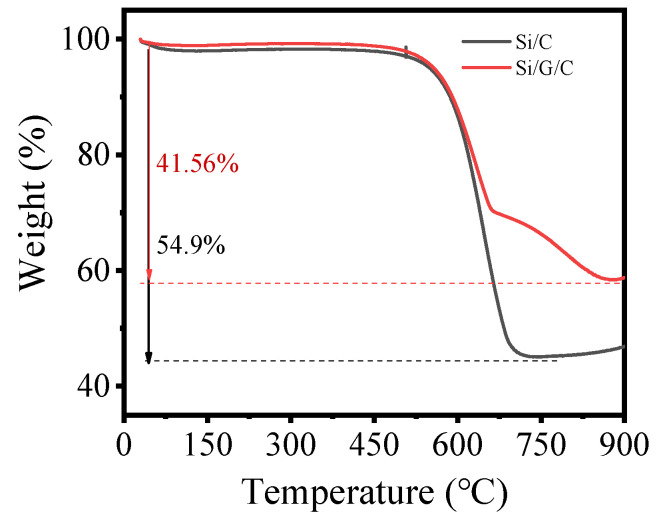
TG profiles.

**Figure 3 molecules-28-00464-f003:**
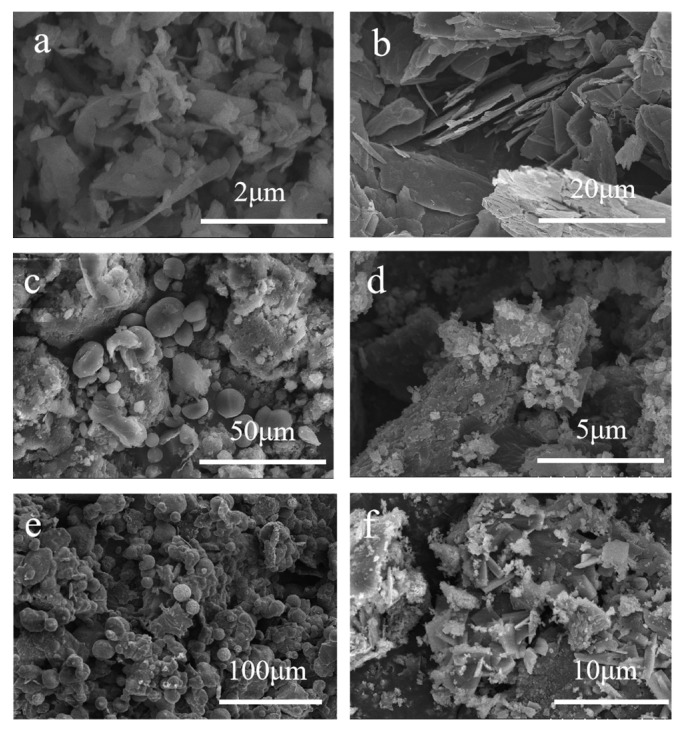
SEM images of (**a**) Si, (**b**) Graphite, (**c**) Si/C composite precursor, (**d**) Si/C, (**e**) Si/G/C composite precursor and (**f**) Si/G/C materials.

**Figure 4 molecules-28-00464-f004:**
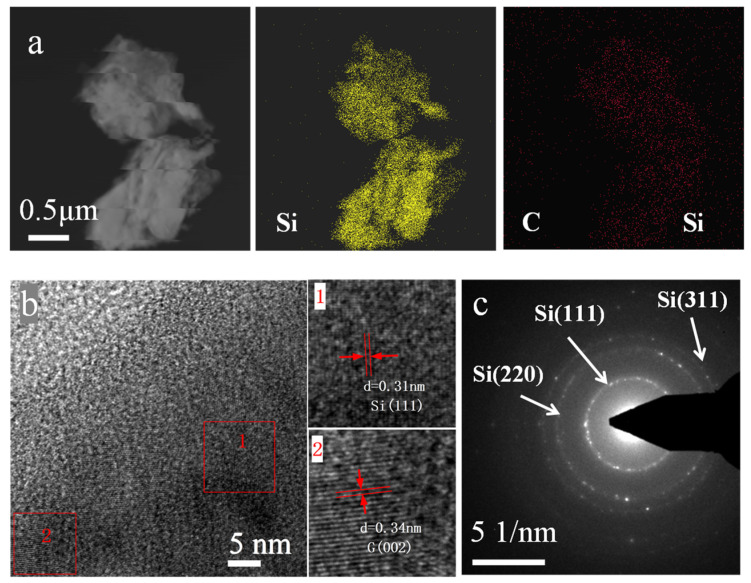
(**a**) EDS elemental mapping (Si, C), (**b**) HRTEM, and (**c**) SAED pattern images of Si/G/C composite.

**Figure 5 molecules-28-00464-f005:**
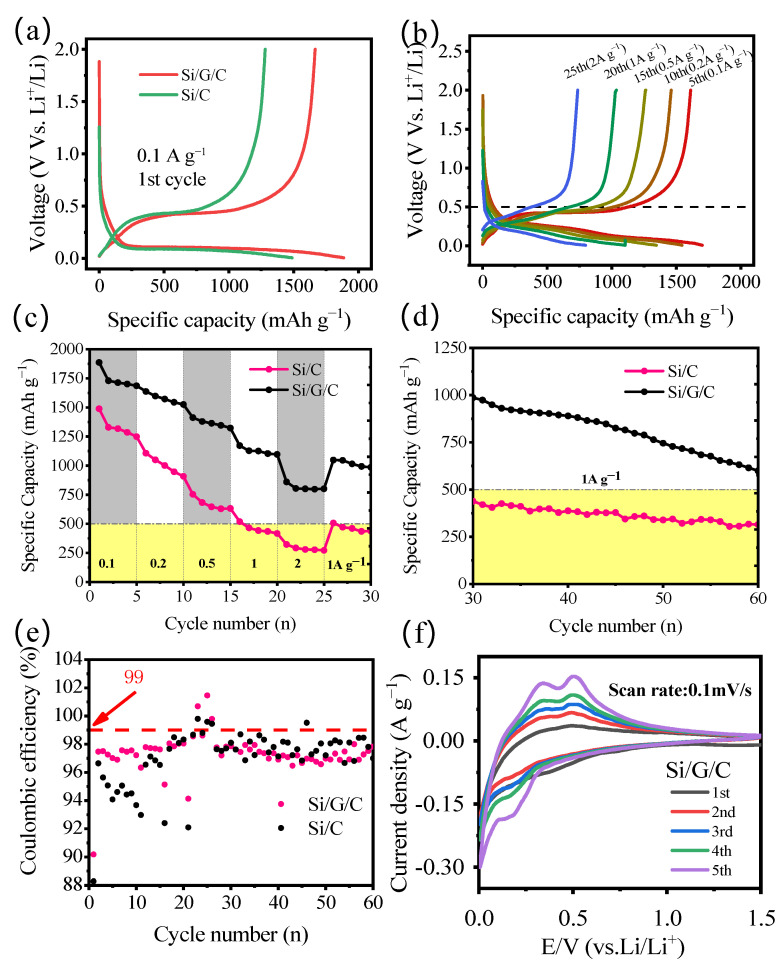
(**a**) The initial charge and discharge profiles of Si/G/C and Si/C anodes at 0.1 A g^−1^. (**b**) Charge/discharge profiles of the Si/G/C anode under various current densities. (**c**) Rate capabilities of Si/G/C and Si/C anodes measured under various current densities from 0.1 A g^−1^ to 2 A g^−1^. (**d**) The cycling performance of the Si/G/C and Si/C anodes at a current density of 1 A g^−1^. (**e**) The coulombic efficiency of Si/G/C. (**f**) Cycle voltammetry of the Si/G/C composite at different cycles in the potential range from 0.01 to 1.5 V (vs. Li/Li^+^).

## Data Availability

The data presented in this study is available in Appendix A.

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
