# Peer review of "Silicon/Graphite/Amorphous Carbon as Anode Materials for Lithium Secondary Batteries"

_molecules, 2023, doi:10.3390/molecules28020464_

Round 1
Reviewer 1 Report
The present manuscript by Duan et al. titled “Silicon/Graphite/Amorphous Carbon as Anode Materials for Lithium Secondary Batteries” is a timely contribution to the molecules journal due to the high technical relevance of the treated research area and the high quality of the work. The authors have synthetized promising anode materials based on silicon and carbon (graphitic modification and amorphous carbon). They have characterized the samples using X-ray diffraction, scanning electron microscopy, thermogravimetry and electrochemical measurements like current densities and cyclovoltammetry curves. The authors found very favorable properties of the composite material like for instance high (initial) discharge capacity and coulombic efficiency.
Overall, the details on the experimental setups and conditions (room temperature measurements) are sufficient to guarantee repeatability by other workers. The quality of figures is high and the language clear. The structure as well as length of the manuscript are appropriate. Based on that I am inclined to recommend publication of the present manuscript as it stands.
Author Response
We would like to thank you for reading our paper carefully and giving the above positive comments. Thank you for your affirmation.We hope this article can be included as soon as possible.
Reviewer 2 Report
In this manuscript, the authors synthesized silicon/graphite/amorphous carbon (Si/G/C) composites by a facile spray drying method followed by heating treatment. In addition, the Si/G/C composites were used as anode materials for lithium-ion batteries (LIBs). The authors intended to examine the effect of graphite of Si/G/C composites on the performance of LIBs. This manuscript was written well and presented fair electrochemical and physicochemical characterizations. However, following revisions should be made before publication.
1. There are some typographical errors that should be corrected, such as superscript, significant figures, unit and so on.
2. N2 adsorption–desorption isotherms, specific surface area, and pore size distribution of Si/G/C and Si/C should be provided.
3. Some electrochemical data should be supplemented, including the first five CV curves of Si/C, CV curves at different scan rates, log(i)–log(v) plots, electrochemical impedance spectra, fitting Z′ and ω−1/2, GITT profiles, and Li+ diffusion coefficients of Si/G/C and Si/C.
4. The cycle number should be reach at least 100.
5. In the introduction section, some articles about silicon/carbon-based materials should be added: doi.org/10.3390/nano12162875.
Reviewer 3 Report
This manuscript reports a novel Si/G/C composite anode for lithium-ions batteries by a spray drying method. The graphite and amorphous carbon in the composite act as a buffer matrix for the Li-Si alloy to alleviate the volume fluctuation during cycling. Furthermore, amorphous carbon connected the Si and graphite successfully, generating a conductive network. As a result, the Si/G/C composite exhibits a superior electrochemical performance. The manuscript is well organized and the date are solid. However, some points need to be further clarified and the grammatical errors should be further checked throughout the manuscript. I think the manuscript has merit but requires minor revision before it can be accepted for publication. Suggestion and question in following are helpful to improve manuscript quality.
1. Some comments to figures:
1) A high-quality picture should be provided in Figure 4a.
2) The annotation in Figure 2 is not consistent with the manuscript. Please check and correct it carefully.
3) The symbol of lithium ion should be corrected in Figure 5f.
2. Please detail some important parameters in experimental section, such as the molecular weight of PVP.
3. Why use spray drying instead of other methods in this manuscript, such as freeze-drying or blast drying?
4. The CV and charge/discharge curves of Si/G/C are encouraged to discuss in detail, for example, it is a reversible two steps reaction (conversion and alloy) or only one step?
5. What role does PVP play throughout the experiment?
Author Response
Point 1: Some comments to figures:
1) A high-quality picture should be provided in Figure 4a.
2) The annotation in Figure 2 is not consistent with the manuscript. Please check and correct it carefully.
3) The symbol of lithium ion should be corrected in Figure 5f.
Response 1: Sorry for the mistakes. We have corrected them.
Point 2: Please detail some important parameters in experimental section, such as the molecular weight of PVP.
Response 2: The molecular weight of PVP (360,000)has been added.
Point 3: Why use spray drying instead of other methods in this manuscript, such as freeze-drying or blast drying?
Response 3: Spray drying relies on heat and generally involves temperatures of 80°C or higher. It’s often a more cost effective alternative to other drying methods such as freeze drying and blast drying and and offers similar results and the capacity to meet the demands of higher throughput laboratories.
Cost is one of the biggest factors considered when choosing between freeze drying and spray drying. Generally, spray drying is a cheaper, faster and more energy efficient alternative to freeze drying.
Point 4: The CV and charge/discharge curves of Si/G/C are encouraged to discuss in detail, for example, it is a reversible two steps reaction (conversion and alloy) or only one step?
Response 4: Thank you for your valuable suggestion. In our study, the conversion reaction is partially reversible, and the alloying reaction is reversible. Some discussion will add in article.
Point 5: What role does PVP play throughout the experiment?
Response 5: The addition of PVP is beneficial to prevent the sedimentation and agglomeration of solution’s particles such as Si and graphite.
Round 2
Reviewer 2 Report
The questions posed by the reviewer were well answered, and the electrochemical and physicochemical characterization data were supplemented. This manuscript can be accepted now.